# Property-Aware Relation Networks for Few-Shot Molecular Property Prediction

**Yaqing Wang**[1][*]   **Abulikemu Abuduweili**[1,2][*]   **Quanming Yao**[3][†]   **Dejing Dou**[1]

[1]Baidu Research, Baidu Inc., China
[2]The Robotics Institute, Carnegie Mellon University, USA
[3]Department of EE, Tsinghua University, China
{wangyaqing01, v_abuduweili, doudejing}@baidu.com
qyaoaa@tsinghua.edu.cn

## Abstract

Molecular property prediction plays a fundamental role in drug discovery to identify candidate molecules with target properties. However, molecular property prediction is essentially a few-shot problem, which makes it hard to use regular machine learning models. In this paper, we propose Property-Aware Relation networks (PAR) to handle this problem. In comparison to existing works, we leverage the fact that both relevant substructures and relationships among molecules change across different molecular properties. We first introduce a property-aware embedding function to transform the generic molecular embeddings to substructure-aware space relevant to the target property. Further, we design an adaptive relation graph learning module to jointly estimate molecular relation graph and refine molecular embeddings w.r.t. the target property, such that the limited labels can be effectively propagated among similar molecules. We adopt a meta-learning strategy where the parameters are selectively updated within tasks in order to model generic and property-aware knowledge separately. Extensive experiments on benchmark molecular property prediction datasets show that PAR consistently outperforms existing methods and can obtain property-aware molecular embeddings and model molecular relation graph properly.

## 1   Introduction

Drug discovery is an important biomedical task, which targets at finding new potential medical compounds with desired properties such as better absorption, distribution, metabolism, and excretion (ADME), low toxicity and active pharmacological activity [1, 2, 3]. It is recorded that drug discovery takes more than 2 billion and at least 10 years in average while the clinical success rate is around 10% [4, 5, 6]. To speed up this process, quantitative structure property/activity relationship (QSPR/QSAR) modeling uses machine learning methods to establish the connection between molecular structure and particular properties [7]. It usually consists of two components: a molecular encoder which encodes molecular structure as a fixed-length molecular representation, and a predictor which estimates the activity of a certain property based on the molecular representation. Predictive models can be leveraged in virtual screening to discover potential molecules more efficiently [8]. However, molecular property prediction is essentially a few-shot problem, which makes it hard to solve. Only a small amount of candidate molecules can pass virtual screening to be evaluated in the lead optimization stage of drug discovery [9]. After a series of wet-lab experiments, most candidates

---

[*]Equal contribution. A. Abuduweili did his work during internship at Baidu Research.
[†]Correspondence to.

35th Conference on Neural Information Processing Systems (NeurIPS 2021).

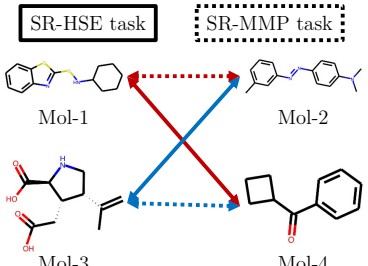

| | Molecule | | Label | |
|---|---|---|---|---|
| ID | SMILES | | SR-HSE | SR-MMP |
| Mol-1 | c1ccc2sc(SNC3CCCCC3)nc2c1 | | 1 | 1 |
| Mol-2 | Cc1cccc(/N=N/c2ccc(N(C)C)cc2)c1 | | 0 | 1 |
| Mol-3 | C=C(C)[C@H]1CN[C@H](C(=O)O)[C@H]1CC(=O)O | | 0 | 0 |
| Mol-4 | O=C(c1ccccc1) C1CCC1 | | 1 | 0 |

Figure 1: Examples of relation graphs for the same molecules coexisting in two tasks of Tox21. Red (blue) edges mean the connected molecules are both active (inactive) on the target property.

eventually fail to be a potential drug due to the lack of any desired properties [7]. These together result in a limited number of labeled data [10].

Few-shot learning (FSL) [11, 12] methods target at generalizing from a limited number of labeled data. Recently, they have also been introduced into molecular property prediction [3, 8]. These methods attempt to learn a predictor from a set of property prediction tasks and generalize to predict new properties given a few labeled molecules. As molecules can be naturally represented as graphs, graph-based molecular representation learning methods use graph neural networks (GNNs) [13, 14] to obtain graph-level representation as the molecular embedding. Specifically, the pioneering IterRefLSTM [3] adopts GNN as the molecular encoder and adapts a classic FSL method [15] proposed for image classification to handle few-shot molecular prediction tasks. The recent Meta-MGNN [8] leverages a GNN pretrained from large-scale self-supervised tasks as molecular encoder and introduces additional self-supervised tasks such as bond reconstruction and atom type prediction to be jointly optimized with the molecular property prediction tasks.

However, aforementioned methods neglect two key facts in molecular property prediction. The first fact is that different molecular properties are attributed to different molecular substructures as found by previous QSPR studies [16, 17, 18]. However, IterRefLSTM and Meta-MGNN use graph-based molecular encoder to encode molecules regardless of target properties whose relevant substructures are quite different. The second fact is that the relationship among molecules also vary w.r.t. the target property. This can be commonly observed in benchmark molecular property prediction datasets. As shown in Figure 1, Mol-1 and Mol-4 from the Tox21 dataset [19] have the same activity in SR-HSE task while acting differently in SR-MMP task. However, existing works fail to leverage such relation graph among molecules.

To handle these problems, we propose Property-Aware Relation networks (PAR) which is compatible with existing graph-based molecular encoders, and is further equipped with the ability to obtain property-aware molecular embeddings and model molecular relation graph adaptively. Specifically, our contribution can be summarized as follows:

- We propose a property-aware embedding function which co-adapts each molecular embedding with respect to context information of the task and further projects it to a substructure-aware space w.r.t. the target property.

- We propose an adaptive relation graph learning module to jointly estimate molecular relation graph and refine molecular embeddings w.r.t. the target property, such that the limited labels can be effectively propagated among similar molecules.

- We propose a meta-learning strategy to selectively update parameters within each task, which is particularly helpful to separately capture the generic knowledge shared across different tasks and those specific to each property prediction task.

- We conduct extensive empirical studies on real molecular property prediction datasets. Results show that PAR consistently outperforms the others. Further model analysis shows PAR can obtain property-aware molecular embeddings and model molecular relation graph properly.

**Notation.** In the sequel, we denote vectors by lowercase boldface, matrices by uppercase boldface, and sets by uppercase calligraphic font. For a vector $\mathbf{x}$, $[\boldsymbol{x}]_i$ denotes the $i$th element of $\mathbf{x}$. For a matrix $\mathbf{X}$, $[\mathbf{X}]_{i:}$ denotes the vector on its $i$th row, $[\mathbf{X}]_{ij}$ denotes the $(i, j)$th element of $\mathbf{X}$. The superscript $(\cdot)^{\top}$ denotes the matrix transpose.

## 2 Review: Graph Neural Networks (GNNs)

A graph neural network (GNN) can learn expressive node/graph representation from the topological structure and associated features of a graph via neighborhood aggregation [13, 20, 21]. Consider a graph $\mathcal{G} = \{\mathcal{V}, \mathcal{E}\}$ with node feature $\mathbf{h}_v^{(0)}$ for each node $v \in \mathcal{V}$ and edge feature $\mathbf{b}_{vu}$ for each edge $e_{vu} \in \mathcal{E}$ between nodes $v, u$. At the $l$th layer, GNN updates the node embedding $\mathbf{h}_v^{(l)}$ of node $v$ as:

$$\mathbf{h}_v^{(l)} = \texttt{UPDATE}^{(l)} \left( \mathbf{h}_v^{(l-1)}, \texttt{AGGREGATE}^{(l)} \left( \{(\mathbf{h}_v^{(l-1)}, \mathbf{h}_u^{(l-1)}, \mathbf{b}_{vu}) | u \in \mathcal{N}(v)\} \right) \right), \qquad (1)$$

where $\mathcal{N}(v)$ is a set of neighbors of $v$. After $L$ iterations of aggregation, the graph-level representation $\mathbf{g}$ for $\mathcal{G}$ is obtained as

$$\mathbf{g} = \texttt{READOUT} \left( \{\mathbf{h}_v^{(L)} | v \in \mathcal{V}\} \right), \qquad (2)$$

where $\texttt{READOUT}(\cdot)$ function aggregates all node embeddings into the graph embedding [22].

Our paper is related to GNN in two aspects: (i) use graph-based molecular encoder to obtain molecular representation, and (ii) conduct graph structure learning to model relation graph among molecules.

**Graph-based Molecular Representation Learning.** Representing molecules properly as fixed-length vectors is vital to the success of downstream biomedical applications [23]. Recently, graph-based molecular representation learning methods are popularly used and obtain state-of-the-art performance. A molecule $\mathbf{x}_i$ is represented as an undirected graph $\mathcal{G}_i = \{\mathcal{V}_i, \mathcal{E}_i\}$, where each node $v \in \mathcal{V}_i$ represents an atom with feature $\mathbf{h}_v^{(0)} \in \mathbb{R}^{d^n}$ and each edge $e_{vu} \in \mathcal{E}_i$ represents the bond between two nodes $v, u$ with feature $\mathbf{b}_{vu} \in \mathbb{R}^{d^e}$. Graph-based molecular representation learning methods use GNNs to obtain graph-level representation $\mathbf{g}_i$ as molecular embedding. Examples include graph convolutional networks (GCN) [24], graph attention networks (GAT) [25], message passing neural networks (MPNN) [20], graph isomorphism network (GIN) [22], pretrained GNN (Pre-GNN) [26] and GROVER [9].

Existing two works in few-shot molecular property prediction both use graph-based molecular encoder to obtain molecular embeddings: IterRefLSTM [3] uses GCN while Meta-MGNN [8] uses Pre-GNN. Using these graph-based molecular encoders cannot discover molecular substructures corresponding to the target property. There exist GNNs which handle subgraphs [27, 28, 29], which are usually predefined or simply K-hop neighborhood. While discovering and enumerating molecular substructures is extremely hard even for domain experts [17, 30]. In this paper, we first obtain molecular embeddings using graph-based molecular encoders. We further learn to extract relevant substructure embeddings w.r.t. the target property upon these generic molecular embeddings, which is more effective and improves the performance.

**Graph Structure Learning.** As the provided graphs may not be optimal, a number of graph structure learning methods target at jointly learning graph structure and node embeddings [31, 32, 33]. In general, they iterate over two procedures: (i) estimate adjacency matrix (i.e., refining neighborhood $u \in \mathcal{N}(v)$) which encodes graph structure from the current node embeddings; and (ii) apply a GNN on this updated graph to obtain new node embeddings.

There exist some FSL methods [34, 35, 36, 37, 38] which learn to construct fully connected relation graph among images in a $N$-way $K$-shot few-shot image classification task. Their methods cannot work for the 2-way $K$-shot property prediction tasks where choosing a wrong neighbor in the different class will heavily deteriorate the quality of molecular embeddings. We share the same spirit of learning relation graph, and further design several regularizations to encourage our adaptive property-aware relation graph learning module to select correct neighbors.

## 3 Proposed Method

In this section, we present the details of PAR, whose overall architecture is shown in Figure 2. Considering few-shot molecular property prediction problem, we first use a specially designed embedding function to obtain property-aware molecular embedding for each molecule, and then adaptively learn relation graph among molecules which allows effective propagation of the limited labels. Finally, we describe our meta-learning strategy to train PAR.

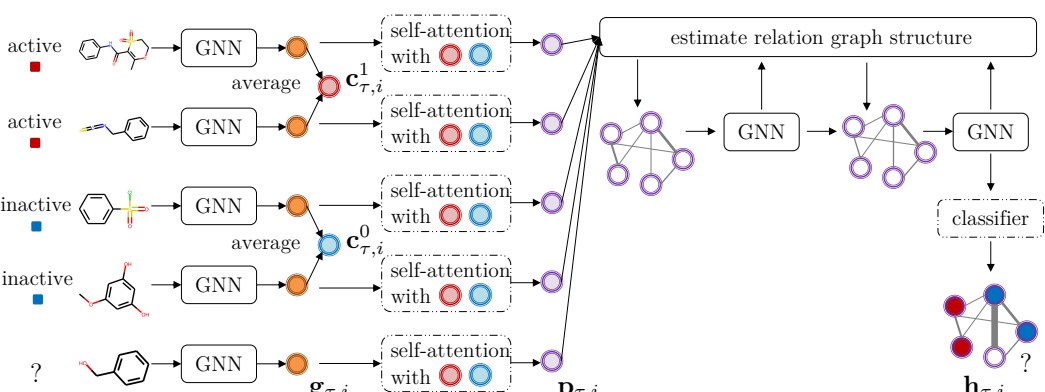

Figure 2: The architecture of the proposed PAR, where we plot a 2-way 2-shot task from Tox21. PAR is optimized over a set of tasks. Within each task $\mathcal{T}_\tau$, the modules with dotted lines are fine-tuned on support set $\mathcal{S}_\tau$ and those with solid lines are fixed. A query molecule $\mathbf{x}_{\tau,i}$ will first be represented as $\mathbf{g}_{\tau,i}$ using graph-based molecular encoder, then transformed to $\mathbf{p}_{\tau,i}$ by our property-aware embedding function. This $\mathbf{p}_{\tau,i}$ further co-adapts with embeddings of molecules in $\mathcal{S}_\tau$ on the relation graph as $\mathbf{h}_{\tau,i}$, which is taken as the final molecular embedding and used for class prediction.

### 3.1 Problem Definition

Following the problem definition adopted by IterRefLSTM [3] and Meta-MGNN [8], the target is to learn a predictor from a set of few-shot molecular property prediction tasks $\{\mathcal{T}_\tau\}_{\tau=1}^{N_t}$ and generalize to predict new properties given a few labeled molecules. The $\tau$th task $\mathcal{T}_\tau$ predicts whether a molecule $\mathbf{x}_{\tau,i}$ with index $i$ is active ($y_{\tau,i} = 1$) or inactive ($y_{\tau,i} = 0$) on a target property, provided with a small number of $K$ labeled samples per class. This $\mathcal{T}_\tau$ is then formulated as a 2-way $K$-shot classification task with a support set $\mathcal{S}_\tau = \{(\mathbf{x}_{\tau,i}, y_{\tau,i})\}_{i=1}^{2K}$ containing the $2K$ labeled samples and a query set $\mathcal{Q}_\tau = \{(\mathbf{x}_{\tau,j}, y_{\tau,j})\}_{j=1}^{N_\tau^q}$ containing $N_\tau^q$ unlabeled samples to be classified.

### 3.2 Property-aware Molecular Embedding

As different molecular properties are attributed to different molecule substructures, we design a property-aware embedding function to transform the generic molecular embeddings to substructure-aware space relevant to the target property.

As introduced in Section 2, graph-based molecular encoders can obtain good molecular embeddings. By learning from large-scale tasks, they can capture generic information shared by molecules [26, 9]. Thus, we first use a graph-based molecular encoder such as GIN [22] and Pre-GNN [26] to extract a molecular embedding $\mathbf{g}_{\tau,i} \in \mathbb{R}^{d^g}$ of length $d^g$ for each $\mathbf{x}_{\tau,i}$. The parameter of this graph-based molecular encoder is denoted as $\mathbf{W}_g$.

However, existing graph-based molecular encoders cannot capture property-aware substructures. Especially when learning across tasks, a molecule can be evaluated for multiple properties. This leads to a one-to-many relationship between a molecule and properties, which makes few-shot molecular property prediction particularly hard. Thus, we are motivated to implicitly capture substructures in the embedding space w.r.t. the target property of $\mathcal{T}_\tau$. Let $\mathbf{c}_\tau^c$ denote the class prototype for class $c \in \{0, 1\}$, which is computed as

$$\mathbf{c}_\tau^c = 1/|\mathcal{S}_\tau^c| \sum_{(\mathbf{x}_{\tau,i}, y_{\tau,i}) \in \mathcal{S}_\tau^c} \mathbf{g}_{\tau,i}, \tag{3}$$

where $\mathcal{S}_\tau^c = \{(\mathbf{x}_{\tau,i}, y_{\tau,i}) | (\mathbf{x}_{\tau,i}, y_{\tau,i}) \in \mathcal{S}_\tau \text{ and } y_{\tau,i} = c\}$. We take these class prototypes as the context information of $\mathcal{T}_\tau$, and further encode them as

$$\mathbf{b}_{\tau,i} = \left[\texttt{softmax}(\mathbf{C}_{\tau,i} \mathbf{C}_{\tau,i}^\top / \sqrt{d^g}) \mathbf{C}_{\tau,i}\right]_{1:} \text{ with } \mathbf{C}_{\tau,i}^\top = [\mathbf{g}_{\tau,i}, \mathbf{c}_\tau^0, \mathbf{c}_\tau^1] \in \mathbb{R}^{d^g \times 3}, \tag{4}$$

where $[\cdot]_{j:}$ extracts the $j$th row vector which corresponds to $\mathbf{x}_{\tau,i}$. Here $\mathbf{b}_{\tau,i}$ is computed using scaled dot-product self-attention [39], such that each $\mathbf{g}_{\tau,i}$ can be compared with class prototypes in a dimensional wise manner. The property-aware molecular embedding $\mathbf{p}_{\tau,i}$ is then obtained as

$$\mathbf{p}_{\tau,i} = \texttt{MLP}_{\mathbf{W}_p}(\texttt{concat}[\mathbf{g}_{\tau,i}, \mathbf{b}_{\tau,i}]). \tag{5}$$

$\texttt{MLP}_{\mathbf{W}_p}$ denotes the multilayer perceptron (MLP) parameterized by $\mathbf{W}_p$, which is used to find a lower dimensional space which encodes substructures that are more relevant to the target property of $\mathcal{T}_\tau$. This contextualized $\mathbf{p}_{\tau,i}$ is property-aware which can be more predictive of the target property.

## 3.3 Adaptive Relation Graph Among Molecules

Apart from relevant substructures, the relationship among molecules also changes across properties. As shown in Figure 1, two molecules with a shared property can be different from each other on another property [1, 40, 41]. Therefore, we further propose an adaptive relation graph learning module to capture and leverage this property-aware relation graph among molecules, such that the limited labels can be efficiently propagated between similar molecules.

In this relation graph learning module, we alternately estimate the adjacency matrix of the relation graph among molecules and refine the molecular embeddings on the learned relation graph for $T$ times.

At the $t$th iteration, let $\mathcal{G}_\tau^{(t)}$ denotes the relation graph where $\mathcal{V}_\tau$ takes the $2K$ molecules in $\mathcal{S}_\tau$ and a query molecule in $\mathcal{Q}_\tau$ as nodes. $\mathbf{A}_\tau^{(t)} \in \mathbb{R}^{(2K+1)\times(2K+1)}$ denotes the corresponding adjacency matrix encoding the $\mathcal{G}_\tau^{(t)}$, where $[\mathbf{A}_\tau^{(t)}]_{ij} \geq 0$ if nodes $\mathbf{x}_{\tau,i}, \mathbf{x}_{\tau,j} \in \mathcal{V}_\tau$ are connected. Ideally, the similarity between property-aware molecular embeddings $\mathbf{p}_{\tau,i}, \mathbf{p}_{\tau,j}$ of $\mathbf{x}_{\tau,i}, \mathbf{x}_{\tau,j}$ reveals their relationship under the current property prediction task. Therefore, we set $\mathbf{h}_{\tau,i}^{(0)} = \mathbf{p}_{\tau,i}$ initially.

We first estimate $\mathbf{A}_\tau^{(t)}$ using the current molecular embeddings. The $(i,j)$th element of $[\mathbf{A}_\tau^{(t)}]_{ij}$ records the similarity between $\mathbf{x}_{\tau,i}, \mathbf{x}_{\tau,j}$ which is calculated as:

$$\left[\mathbf{A}_\tau^{(t)}\right]_{ij} = \texttt{MLP}_{\mathbf{W}_a}\left(\exp(-|\mathbf{h}_{\tau,i}^{(t-1)} - \mathbf{h}_{\tau,j}^{(t-1)}|)\right), \tag{6}$$

where $\mathbf{W}_a$ is the parameter of this MLP. The resultant $\mathbf{A}_\tau^{(t)}$ is a dense matrix, which encodes a fully connected $\mathcal{G}_\tau^{(t)}$.

However, a query molecule only has $K$ real neighbors in $\mathcal{G}_\tau^{(t)}$ in a 2-way $K$-shot task. For binary classification, choosing a wrong neighbor in the opposite class will heavily deteriorate the quality of molecular embeddings, especially when only one labeled molecule is provided per class. To avoid the interference of wrong neighbors, we further reduce $\mathcal{G}_\tau^{(t)}$ to a $K$-nearest neighbor ($K$NN) graph, where $K$ is set to be exactly the same as the number of labeled molecules per class in $\mathcal{S}$. The indices of the top $K$ largest $[\mathbf{A}_\tau^{(t)}]_{ij}, j = 1, \ldots, 2K - 1$ for $\mathbf{x}_{\tau,i}$ is recorded in $\mathcal{N}^{(t)}(\mathbf{x}_{\tau,i})$. Then, we set

$$\left[\hat{\mathbf{A}}_\tau^{(t)}\right]_{ij} = \begin{cases} [\mathbf{A}_\tau^{(t)}]_{ij} & \text{if } \mathbf{x}_{\tau,j} \in \mathcal{N}^{(t)}(\mathbf{x}_{\tau,i}) \\ 0 & \text{otherwise} \end{cases}. \tag{7}$$

The values in $[\hat{\mathbf{A}}_\tau^{(t)}]$ are normalized to range between 0 and 1, which is done by applying softmax function on each row $[\hat{\mathbf{A}}_\tau^{(t)}]_{i:}$. This normalization can also be done by z-score, min-max and sigmoid normalization. Then, we co-adapt each node embedding $\mathbf{h}^{(t)}$ with respect to other node embeddings on this updated relation graph encoded $\hat{\mathbf{A}}_\tau^{(t)}$. Let $\mathbf{H}_\tau^{(t)}$ denote all node embeddings collectively where the $i$th row corresponds to $\mathbf{h}_{\tau,i}^{(t)}$. $\mathbf{H}_\tau^{(t)}$ is updated as

$$\mathbf{H}_\tau^{(t)} = \texttt{LeakyReLu}(\hat{\mathbf{A}}_\tau^{(t)}\mathbf{H}_\tau^{(t-1)}\mathbf{W}_r), \tag{8}$$

where $\mathbf{W}_r$ is a learnable parameter.

After $T$ iterations, we return $\mathbf{h}_{\tau,i} = [\mathbf{H}_\tau^{(T)}]_{i:}$ as the final molecular embedding for $\mathbf{x}_{\tau,i}$, and $\hat{\mathbf{A}}_\tau = \hat{\mathbf{A}}_\tau^{(T)}$ as the final optimized relation graph.

Denote $\hat{\mathbf{y}}_{\tau,i}$ as the class prediction of $\mathbf{x}_{\tau,i}$ w.r.t. active/inactive, which is calculated as

$$\hat{\mathbf{y}}_{\tau,i} = \texttt{softmax}(\mathbf{W}_c \cdot \mathbf{h}_{\tau,i}), \tag{9}$$

where $[\texttt{softmax}(\mathbf{x})]_i = \exp([\mathbf{x}]_i)/\sum_j^{2K+1}\exp([\mathbf{x}]_j)$ is applied per row, and $\mathbf{W}_c$ is a parameter.

**Algorithm 1** Meta-training procedure for PAR.

---

1: initialize $\boldsymbol{\theta} = \{\mathbf{W}_g, \mathbf{W}_a, \mathbf{W}_r\}$ and $\boldsymbol{\Phi} = \{\mathbf{W}_p, \mathbf{W}_c\}$ randomly; if a pretrained molecular encoder is available, take its parameter as $\mathbf{W}_g$;
2: **while** not done **do**
3:     sample a batch of tasks $\mathcal{T}_\tau$;
4:     **for** all $\mathcal{T}_\tau$ **do**
5:         sample support set $\mathcal{S}_\tau$ and query set $\mathcal{Q}_\tau$ from $\mathcal{T}_\tau$;
6:         obtain molecular embedding $\mathbf{g}_{\tau,i}$ for each $\mathbf{x}_{\tau,i}$ by a graph-based molecular encoder;
7:         adapt $\mathbf{g}_{\tau,i}$ to be property-aware $\mathbf{p}_{\tau,i}$ by (5);
8:         initialize node embeddings as $\mathbf{h}_{\tau,i}^{(0)} = \mathbf{p}_{\tau,i}$;
9:         **for** $t = 1, \ldots, T$ **do**
10:            estimate adjacency matrix $\mathbf{A}_\tau^{(t)}$ of relation graph among molecules using $\mathbf{h}_{\tau,i}^{(t-1)}$ by (6);
11:            refine $\mathbf{h}_{\tau,i}^{(t)}$ on the updated relation graph $\mathbf{A}_\tau^{(t)}$ by (8);
12:         **end for**
13:         obtain class prediction $\hat{\mathbf{y}}_{\tau,i}$ using $\mathbf{h}_{\tau,i} = \mathbf{h}_{\tau,i}^{(T)}$;
14:         evaluate training loss $\mathcal{L}(\mathcal{S}_\tau, f_{\boldsymbol{\theta},\boldsymbol{\Phi}})$ on $\mathcal{S}_\tau$;
15:         fine-tune $\boldsymbol{\Phi}$ as $\boldsymbol{\Phi}_\tau$ by (11);
16:         evaluate testing loss $\mathcal{L}(\mathcal{Q}_\tau, f_{\boldsymbol{\theta},\boldsymbol{\Phi}_\tau})$ on $\mathcal{Q}_\tau$;
17:     **end for**
18:     update $\boldsymbol{\theta}$ and $\boldsymbol{\Phi}$ by (12);
19: **end while**

---

### 3.4 Training and Inference

We denote PAR as $f_{\boldsymbol{\theta},\boldsymbol{\Phi}}$. In particular, $\boldsymbol{\theta} = \{\mathbf{W}_g, \mathbf{W}_a, \mathbf{W}_r\}$ denotes the collection of parameters of graph-based molecular encoder and adaptive relation graph learning module. While $\boldsymbol{\Phi} = \{\mathbf{W}_p, \mathbf{W}_c\}$ includes the parameters of property-aware molecular embedding function and classifier.

We adopt the gradient-based meta-learning strategy [42]: a good initialized parameter is learned from a set of meta-training tasks $\{\mathcal{T}_\tau\}_{\tau=1}^{N_t}$, which acts as starting point for each task $\mathcal{T}_\tau$. Upon this general strategy, we selectively update parameters within tasks in order to encourage the model to capture generic and property-aware information separately. In detail, we keep $\boldsymbol{\theta}$ fixed while fine-tuning $\boldsymbol{\Phi}$ as $\boldsymbol{\Phi}_\tau$ on $\mathcal{S}_\tau$ in each $\mathcal{T}_\tau$. The training loss $\mathcal{L}(\mathcal{S}_\tau, f_{\boldsymbol{\theta},\boldsymbol{\Phi}})$ evaluated on $\mathcal{S}_\tau$ takes the form:

$$\mathcal{L}(\mathcal{S}_\tau, f_{\boldsymbol{\theta},\boldsymbol{\Phi}}) = \sum\nolimits_{(\mathbf{x}_{\tau,i}, y_{\tau,i}) \in \mathcal{S}_\tau} g(\mathbf{x}_{\tau,i}, y_{\tau,i}, f_{\boldsymbol{\theta},\boldsymbol{\Phi}}) \tag{10}$$

with $g(\mathbf{x}_{\tau,i}, y_{\tau,i}, f_{\boldsymbol{\theta},\boldsymbol{\Phi}}) = -\mathbf{y}_{\tau,i}^\top \cdot \log(\hat{\mathbf{y}}_{\tau,i}) + \sum_{(\mathbf{x}_{\tau,m}, y_{\tau,m}) \in \mathcal{S}_\tau} ([\mathbf{A}_\tau^*]_{im} - [\hat{\mathbf{A}}_\tau]_{im})^2$ where $\mathbf{y}_{\tau,i} \in \mathbb{R}^2$ is a one-hot vector with all 0s but a single one denoting the index of the ground-truth class $c \in \{0, 1\}$, and $\mathbf{A}_\tau^*$ is computed using ground-truth labels with $[\mathbf{A}_\tau^*]_{ij} = 1$ if $y_{\tau,i} = y_{\tau,j}$ and 0 otherwise. The first term is the cross entropy for classification. The second term is the specially designed neighbor alignment regularizer which penalizes the selection of wrong neighbors in the relation graph.

$\boldsymbol{\Phi}_\tau$ is obtained by taking a few gradient descent updates with learning rate $\alpha$:

$$\boldsymbol{\Phi}_\tau = \boldsymbol{\Phi} - \alpha \nabla_{\boldsymbol{\Phi}} \mathcal{L}(\mathcal{S}_\tau, f_{\boldsymbol{\theta},\boldsymbol{\Phi}}). \tag{11}$$

$\boldsymbol{\theta}^*$ and $\boldsymbol{\Phi}^*$ are learned by optimizing the following objective:

$$\min_{\boldsymbol{\theta},\boldsymbol{\Phi}} \sum\nolimits_{\tau=1}^{N_t} \mathcal{L}(\mathcal{Q}_\tau, f_{\boldsymbol{\theta},\boldsymbol{\Phi}_\tau}), \tag{12}$$

where the loss $\mathcal{L}(\mathcal{Q}_\tau, f_{\boldsymbol{\theta},\boldsymbol{\Phi}_\tau})$ is calculated in the same form of (10) but is evaluated on $\mathcal{Q}_\tau$ instead. It is also optimized by gradient descent [42].

The complete algorithm of PAR is shown in Algorithm 1. Line 6-7 correspond to property-aware embedding $\mathbf{p}_{\tau,i}$ which encodes substructure w.r.t the target property (Section 3.2). Line 8-12 correspond to adaptive relation graph learning which facilitates effective label propagation among similar molecules (Section 3.3).

For inference, the generalization ability of PAR is evaluated on the query set $\mathcal{Q}_{\text{new}}$ of each new task $\mathcal{T}_{\text{new}}$ which tests on new property in meta-testing stage. Still, $\boldsymbol{\theta}^*$ is fixed and $\boldsymbol{\Phi}^*$ is fine-tuned on $\mathcal{S}_{\text{new}}$.

# 4 Experiments

We perform experiments on widely used benchmark few-shot molecular property prediction datasets (Table 1) included in MoleculeNet [43]. Details of these benchmarks are in Appendix A.

Table 1: Summary of datasets used.

| Dataset | Tox21 | SIDER | MUV | ToxCast |
|---|---|---|---|---|
| # Compounds | 8014 | 1427 | 93127 | 8615 |
| # Tasks | 12 | 27 | 17 | 617 |
| # Meta-Training Tasks | 9 | 21 | 12 | 450 |
| # Meta-Testing Tasks | 3 | 6 | 5 | 167 |

## 4.1 Experimental Settings

**Baselines.** In the paper, we compare our **PAR**[3] (Algorithm 1) with two types of baselines: (i) FSL methods with graph-based molecular encoder learned from scratch, including **Siamese** [44], **ProtoNet** [45], **MAML** [42], **TPN** [35], **EGNN** [36], and **IterRefLSTM** [3]; and (ii) methods which leverage pretrained graph-based molecular encoder including **Pre-GNN** [26], **Meta-MGNN** [8], and **Pre-PAR** which is our PAR equipped with Pre-GNN. We use results of Siamese and IterRefLSTM reported in [3] as the codes are not available. For the other methods, we implement them using public codes of the respective authors. More implementation details are in Appendix B.

**Generic Graph-based Molecular Representation.** Following [26, 8], we use RDKit [46] to build molecular graphs from raw SMILES, and to extract atom features (atom number and chirality tag) and bond features (bond type and bond direction). For all methods re-implemented by us, we use GIN [22] as the graph-based molecular encoder to extract molecular embeddings. Pre-GNN, Meta-MGNN and Pre-PAR further use the pretrained GIN which is also provided by the authors of [26].

**Evaluation Metrics.** Following [26, 8], we evaluate the binary classification performance by ROC-AUC scores calculated on the query set of each meta-testing task. We run experiments for ten times with different random seeds, and report the mean and standard deviations of ROC-AUC computed over all meta-testing tasks.

## 4.2 Performance Comparison

Table 2 shows the results. Results of Siamese, IterRefLSTM and Meta-MGNN on ToxCast are not provided: the first two methods lack codes and are not evaluated on ToxCast before, while Meta-MGNN runs out of memory as it weighs the contribution of each task among all tasks during meta-training. As can be seen, Pre-PAR consistently obtains the best performance, while PAR obtains the best performance among methods using graph-based molecular encoders learned from scratch. In terms of average improvement, PAR obtains significantly better performance than the best baseline learned from scratch (e.g. EGNN) by 1.59%, and Pre-PAR is better than the best baseline with pretrained molecular encoders (e.g. Meta-MGNN) by 1.49%. Pre-PAR also takes less time and episodes to converge than Meta-MGNN, which is shown in Appendix C.1. In addition, we observe that FSL methods that learn relation graphs (i.e., GNN, TPN, EGNN) obtain better performance than the classic ProtoNet and MAML.

## 4.3 Ablation Study

We further compare Pre-PAR and PAR with the following variants: (i) **w/o P**: w/o applying the property-aware embedding function; (ii) **w/o context in P**: w/o context $b_{\tau,i}$ in equation (5); (iii) **w/o R**: w/o using the adaptive relation graph learning; (iv) **w/ cos-sim in R**: use cosine similarity to obtain the adjacency matrix as $[\mathbf{A}_\tau]_{ij} = \mathbf{p}_{\tau,i}^\top \mathbf{p}_{\tau,j}/(\|\mathbf{p}_{\tau,i}\|_2 \|\mathbf{p}_{\tau,j}\|_2)$, then calculate (7) and (8) as in PAR; (v) **w/o $K$NN in R**: w/o reducing $\mathcal{G}_\tau$ to $K$NN graph; (vi) **w/o reg**: w/o using the neighbor alignment regularizer in equation (10); and (vii) **tune all**: fine-tune all parameters on line 15 of Algorithm 1. Note that these variants follows control variates method. They cover all components of training PAR without overlapping functionalities.

Results on 10-shot tasks are in Figure 3. Again, Pre-PAR obtains better performance than PAR due to a better starting point. PAR and Pre-PAR outperform their variants. The removal of any component leads to significant performance drop. In particular, the performance gain of PAR and Pre-PAR with

---

[3]Codes are available at `https://github.com/tata1661/PAR-NeurIPS21`.

Table 2: ROC-AUC scores on benchmark molecular property prediction datasets. The best results (according to the pairwise t-test with 95% confidence) are highlighted in gray. Methods which use pretrained graph-based molecular encoder are marked in green.

| Method | Tox21 | | SIDER | | MUV | | ToxCast | |
|---|---|---|---|---|---|---|---|---|
| | 10-shot | 1-shot | 10-shot | 1-shot | 10-shot | 1-shot | 10-shot | 1-shot |
| Siamese | $80.40_{(0.35)}$ | $65.00_{(1.58)}$ | $71.10_{(4.32)}$ | $51.43_{(3.31)}$ | $59.96_{(5.13)}$ | $50.00_{(0.17)}$ | - | - |
| ProtoNet | $74.98_{(0.32)}$ | $65.58_{(1.72)}$ | $64.54_{(0.89)}$ | $57.50_{(2.34)}$ | $65.88_{(4.11)}$ | $58.31_{(3.18)}$ | $63.70_{(1.26)}$ | $56.36_{(1.54)}$ |
| MAML | $80.21_{(0.24)}$ | $75.74_{(0.48)}$ | $70.43_{(0.76)}$ | $67.81_{(1.12)}$ | $63.90_{(2.28)}$ | $60.51_{(3.12)}$ | $66.79_{(0.85)}$ | $65.97_{(5.04)}$ |
| TPN | $76.05_{(0.24)}$ | $60.16_{(1.18)}$ | $67.84_{(0.95)}$ | $62.90_{(1.38)}$ | $65.22_{(5.82)}$ | $50.00_{(0.51)}$ | $62.74_{(1.45)}$ | $50.01_{(0.05)}$ |
| EGNN | $81.21_{(0.16)}$ | $79.44_{(0.22)}$ | $72.87_{(0.73)}$ | $70.79_{(0.95)}$ | $65.20_{(2.08)}$ | $62.18_{(1.76)}$ | $63.65_{(1.57)}$ | $61.02_{(1.94)}$ |
| IterRefLSTM | $81.10_{(0.17)}$ | $80.97_{(0.10)}$ | $69.63_{(0.31)}$ | $71.73_{(0.14)}$ | $49.56_{(5.12)}$ | $48.54_{(3.12)}$ | - | - |
| PAR | $82.06_{(0.12)}$ | $80.46_{(0.13)}$ | $74.68_{(0.31)}$ | $71.87_{(0.48)}$ | $66.48_{(2.12)}$ | $64.12_{(1.18)}$ | $69.72_{(1.63)}$ | $67.28_{(2.90)}$ |
| Pre-GNN | $82.14_{(0.08)}$ | $81.68_{(0.09)}$ | $73.96_{(0.08)}$ | $73.24_{(0.12)}$ | $67.14_{(1.58)}$ | $64.51_{(1.45)}$ | $73.68_{(0.74)}$ | $72.90_{(0.84)}$ |
| Meta-MGNN | $82.97_{(0.10)}$ | $82.13_{(0.13)}$ | $75.43_{(0.21)}$ | $73.36_{(0.32)}$ | $68.99_{(1.84)}$ | $65.54_{(2.13)}$ | - | - |
| Pre-PAR | $84.93_{(0.11)}$ | $83.01_{(0.09)}$ | $78.08_{(0.16)}$ | $74.46_{(0.29)}$ | $69.96_{(1.37)}$ | $66.94_{(1.12)}$ | $75.12_{(0.84)}$ | $73.63_{(1.00)}$ |

respect to "w/ cos-sim in R" validates the necessity of learning a similarity function from the data rather than using the fixed cosine similarity. We also try to iterate the estimation of relation graph constructed by cosine similarity, but observe a performance drop given more iterations. Results on 1-shot is put in Appendix C.2 where the observations are consistent.

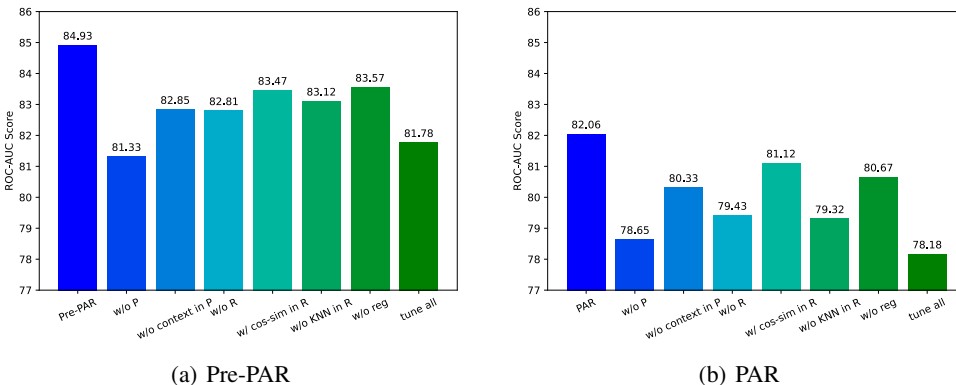

(a) Pre-PAR                    (b) PAR

Figure 3: Ablation study on 10-shot tasks from Tox21.

## 4.4 Using Other Graph-based Molecular Encoders

In the experiments, we use GIN and its pretrained version. However, as introduced in Section 3.2, our PAR is compatible with any existing graph-based molecular encoder introduced in Section 2. Here, we consider the following popular choices as the encoder to output $\mathbf{g}_{\tau,i}$: GIN [22], GCN [24], GraphSAGE [14] and GAT [25], which are either learned from scratch or pretrained. We compare the proposed PAR with simply fine-tuning the encoder on support sets (denote as GNN).

Figure 4 shows the results. As can be seen, GIN is the best graph-based molecular encoder among the four chosen GNNs. PAR outperforms the fine-tuned GNN consistently. This validates the effectiveness of the property-aware molecular embedding function and the adaptive relation graph learning module. We further notice that using pretrained encoders can improve the performance except for GAT, which is also observed in [26].

Although using pretrained graph-based molecular encoders can improve the performance in general, please note that both molecular encoders learned from scratch or pretrained are useful. Pretrained encoders contain rich generic molecular information by learning enormous unlabeled data, while encoders learned from scratch can carry some new insights. For example, the recent DimeNet [47] can model directional information such as bond angles and rotations between atoms, which has no pretrained version. As our proposed method can use any molecular encoder to obtain generic molecular embedding, it can easily accommodate newly proposed molecular encoder w/o or w/ pretraining.

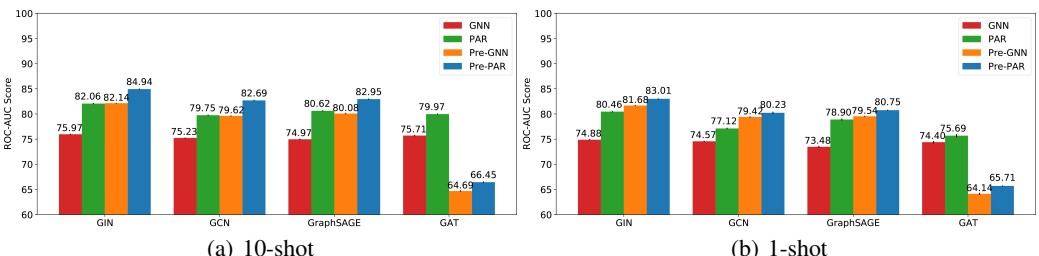

(a) 10-shot

(b) 1-shot

Figure 4: ROC-AUC scores on Tox21 using different graph-based molecular encoders.

## 4.5 Case Study

Finally, we validate whether PAR can obtain different property-aware molecular embeddings and relation graphs for tasks containing overlapping molecules but evaluating different properties.

To examine this under a controlled setting, we sample a fixed group of 10 molecules on Tox21 (Table 5 in Appendix C.3) which coexist in different meta-testing tasks (i.e., the 10th, 11th and 12th tasks). Provided with the meta-learned parameters $\theta^*$ and $\Phi^*$, we take these 10 molecules as the support set to fine-tune $\Phi^*$ as $\Phi^*_\tau$ and keep $\theta^*$ fixed in each task $\mathcal{T}_\tau$. As the support set is fixed now, the ratio of active molecules to inactive molecules among the 10 molecules may not be 1:1 in the three tasks. Thus, the resultant task may not evenly contain $K$ labeled samples per class.

**Visualization of the Learned Relation Graphs.** As described in Section 3.3, PAR returns $\hat{\mathbf{A}}_\tau$ as the adjacency matrix encoding the optimized relation graph among molecules. Each element $[\hat{\mathbf{A}}_\tau]_{ij}$ records the pairwise similarity of the 10 molecules and a random query (which is dropped then). As the number of active and inactive molecules may not be equal in the support set, we no longer reduce adjacency matrices $\mathbf{A}_\tau$ to $\hat{\mathbf{A}}_\tau$ which encodes $K$NN graph. Figure 5 plots the optimized adjacency matrices obtained on all three tasks. As can be observed, PAR obtains different adjacency matrices for different property-prediction tasks. Besides, the learned adjacency matrices are visually similar to the ones computed using ground-truth labels.

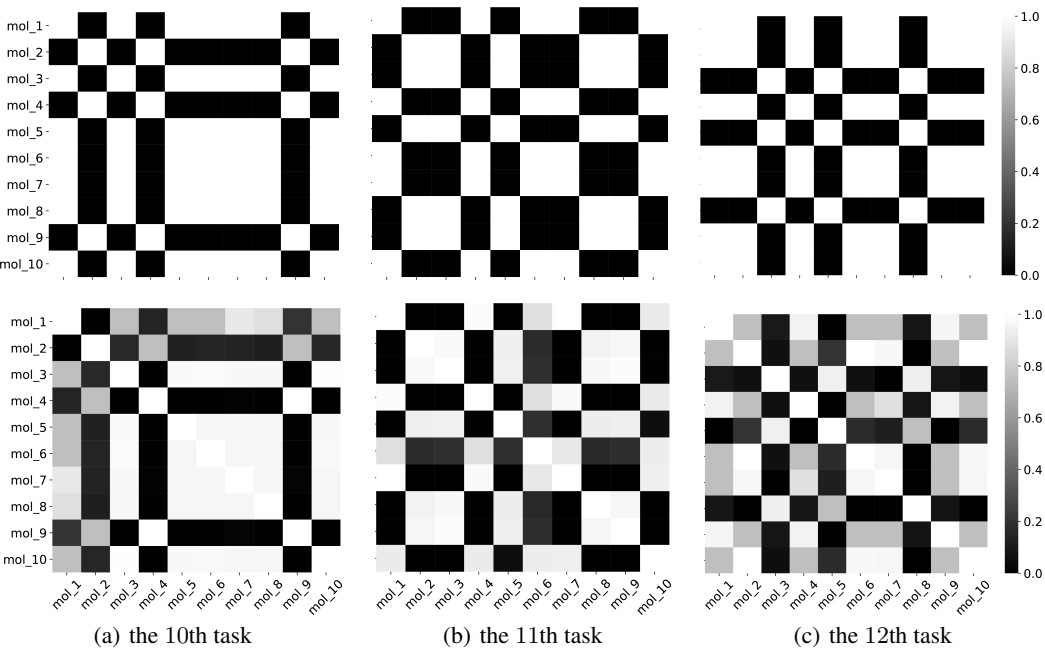

(a) the 10th task

(b) the 11th task

(c) the 12th task

Figure 5: Comparison between $\mathbf{A}^*_\tau$ computed using ground-truth labels (the first row) and adjacency matrix $\mathbf{A}_\tau$ returned by PAR (the second row) for the ten molecules. We set $[\mathbf{A}^*_\tau]_{ij} = 1$ if molecules $\mathbf{x}_{\tau,i}$ and $\mathbf{x}_{\tau,j}$ have the same label and 0 otherwise.

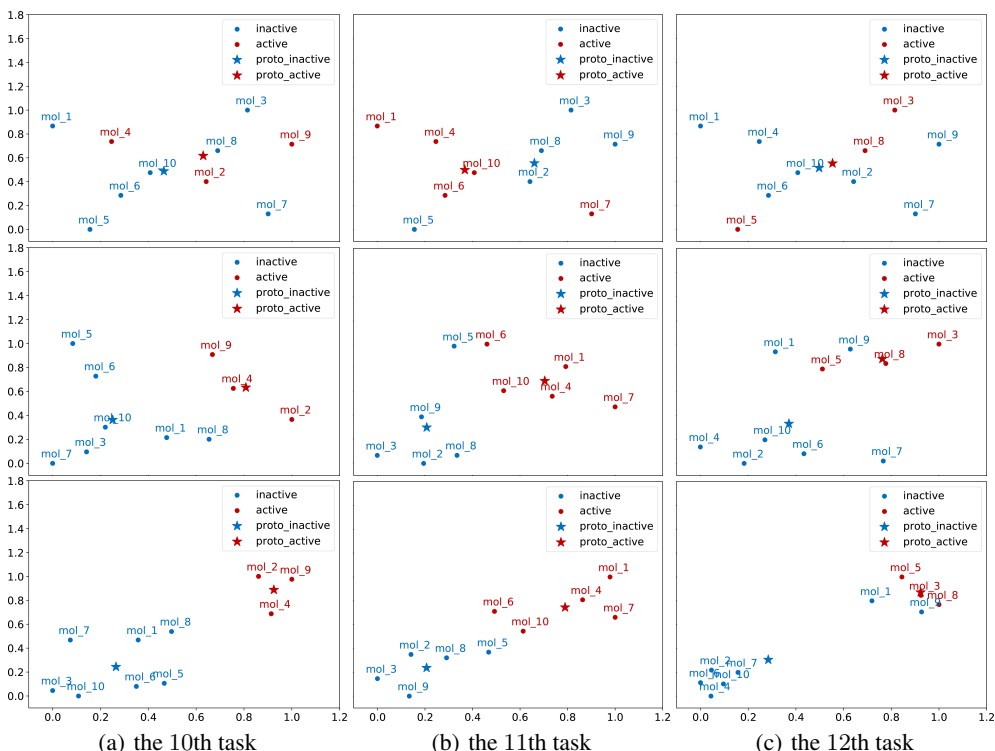

Figure 6: t-SNE visualization of $\mathbf{g}_{\tau,i}$ (the first row), $\mathbf{p}_{\tau,i}$ (the second row), and $\mathbf{h}_{\tau,i}$ (the third row) of the ten molecules. Proto_active (proto_inactive) denotes the class prototype of active (inactive) class.

**Visualization of the Learned Molecular Embeddings.** We also present the t-SNE visualization of $\mathbf{g}_{\tau,i}$ (molecular embedding obtained by graph-based molecular encoders), $\mathbf{p}_{\tau,i}$ (molecular embedding obtained by property-aware embedding function), and $\mathbf{h}_{\tau,i}$ (molecular embedding returned by PAR) for these 10 molecules. For the same $\mathbf{x}_{\tau,i}$, $\mathbf{g}_{\tau,i}$ is the same across 10th, 11th, 12th task, while $\mathbf{p}_{\tau,i}$ and $\mathbf{h}_{\tau,i}$ are property-aware. Figure 6 shows the results. As shown, PAR indeed captures property-aware information during encoding the same molecules for different molecular property prediction tasks. From the first row to the third row in Figure 6, molecular embeddings gradually get closer to the class prototypes on all three tasks.

## 5 Conclusion

We propose Property-Aware Relation networks (PAR) to address the few-shot molecular property prediction problem. PAR contains: a graph-based molecular encoder to encode the topological structure of the molecular graph, atom features, and bond features into a molecular embedding; a property-aware embedding function to obtain property-aware embeddings encoding context information of each task; and an adaptive relation graph learning module to construct a relation graph to effectively propagate information among similar molecules. Empirical results consistently show that PAR obtains state-of-the-art performance on few-shot molecular property prediction problem.

There are several directions to explore in the future. In this paper, PAR is evaluated on biophysics and physiology molecular properties which are modeled as classification tasks. While the prediction of quantum mechanics and physical chemistry properties are mainly regression tasks, it is interesting to extend PAR to handle these different levels of molecular properties. In addition, although PAR targets at few-shot molecular property prediction, the proposed property-aware embedding function, adaptive relation graph learning module, and the neighbor alignment regularizer can be helpful to improve the performance of graph-based molecular encoders in general. Finally, interpreting the substructures learned by PAR is also a meaningful direction.

## Acknowledgements

We sincerely thank the anonymous reviewers for their valuable comments and suggestions. Parts of experiments were carried out on Baidu Data Federation Platform.

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
