# A    Details of Datasets

We perform experiments on widely used benchmark few-shot molecular property prediction datasets[4]:
(i) Tox21 [19] contains assays each measuring the human toxicity of a biological target; (ii)
SIDER [40] records the side effects for compounds used in marketed medicines, where the original
5868 side effect categories are grouped into 27 categories as in [3, 8]; (iii) MUV [1] is designed to
validate virtual screening where active molecules are chosen to be structurally distinct from each
another; and (iv) ToxCast [41] is a collection of compounds with toxicity labels which are obtained
via high-throughput screening. Tox21, SIDER and MUV have public task splits provided by [3],
which are adopted in this paper. For ToxCast, we randomly select 450 tasks for meta-training and use
the rest for meta-testing.

# B    Implementation Details

Experiments are conducted on a 32GB NVIDIA Tesla V100 GPU.

## B.1    Baselines

In the paper, we compare our **PAR** (Algorithm 1) with two types of baselines: (i) FSL methods with
graph-based encoder learned from scratch including **Siamese** [44] which learns dual convolutional
neural networks to identity whether the input molecule pairs are from the same class, **ProtoNet**[5] [45]
which assigns each query molecule with the label of its nearest class prototype, **MAML**[6] [42]
which adapts the meta-learned parameters to new tasks via gradient descent, **TPN**[7] [35] which
conducts label propagation on a relation graph with rescaled edge weight under transductive setting,
**EGNN**[8] [36] which learns to predict edge-labels of relation graph, and **IterRefLSTM** [3] which
adapts Matching Networks [15] to handle molecular property prediction tasks; and (ii) methods
which leverage pretrained graph-based molecular encoder including **Pre-GNN**[9] [26] which pretrains
a GIN [22] using graph-level and node-level self-supervised tasks and is fine-tuned using support
sets, **Meta-MGNN**[10] [8] which uses Pre-GNN as molecular encoder and optimizes the molecular
property prediction task with self-supervised bond reconstruction and atom type predictions tasks,
and **Pre-PAR** which is our PAR equipped with Pre-GNN. GROVER [9] is not compared as it uses a
different set of atom and bond features. We use results of Siamese and IterRefLSTM reported in [3]
as their codes are not available. For the other methods, we implement them using public codes of the
respective authors. We find hyperparameters using the validation set via grid search for all methods.

**Generic Graph-based Molecular Representation.**    For methods re-implemented by us, we use
GIN as the graph-based molecular encoder to extract molecular embeddings in all methods (including
ours). Following [8, 26], we use GIN[11] provided by the authors of [26] which consists of 5 GNN
layers with 300 dimensional hidden units ($d^g = 300$), take average pooling as the READOUT function,
and set dropout rate as 0.5. Pre-GNN, Meta-MGNN and Pre-PAR further use the pretrained GIN
which is also provided by the authors of [26].

## B.2    PAR

In PAR (and Pre-PAR), MLP used in equation (5) and (6) both consist of two fully connected layers
with hidden size 128. We iterate between relation graph estimation and molecular embedding
refinement for two times. We train the model for a maximum number of 2000 episodes. We use
Adam [48] with a learning rate 0.001 for meta training and a learning rate 0.05 for fine-tuning

---

[4]All datasets are downloaded from `http://moleculenet.ai/`.

[5]`https://github.com/jakesnell/prototypical-networks`

[6]We use MAML implemented in learn2learn library at `https://github.com/learnables/learn2learn`.

[7]`https://github.com/csyanbin/TPN-pytorch`

[8]`https://github.com/khy0809/fewshot-egnn`

[9]`http://snap.stanford.edu/gnn-pretrain`

[10]`https://github.com/zhichunguo/Meta-Meta-MGNN`

[11]GIN, GAT, GCN and GraphSAGE and their pretrained versions are obtained from `https://github.com/snap-stanford/pretrain-gnns/`, whose details are in Appendix A of [26].

property-aware molecular embedding function and classifier within each task. We early stop training if the validation loss does not decrease for ten consecutive episodes. Dropout rate is 0.1 except for the graph-based molecular encoder. We summarize the hyperparameters and their range used by PAR in Table 3.

Table 3: Hyperparameters used by PAR.

| Hyperparameter | Range | Selected |
|---|---|---|
| learning rate for fine-tuning $\Phi$ in each task | 0.01~0.5 | 0.05 |
| number of update steps for fine-tuning | 1~5 | 1 |
| learning rate for meta-learning | 0.001 | 0.001 |
| number of layer for MLPs in (5) and (6) | 1~3 | 2 |
| hidden dimension for MLPs in (5) and (6) | 100~300 | 128 |
| dropout rate | 0.0~0.5 | 0.1 |
| hidden dimension for classifier in (9) | 100~200 | 128 |

# C More Experimental Results

## C.1 Computational Cost

Following Table 2, we further compare Pre-PAR with Meta-MGNN in terms of computational cost. We record the training time and training episodes which corresponds to the times of repeating the while loop (line 2-19) in Algorithm 1. The results on 10-shot learning from Tox21 dataset are summarized in Table 4. As shown, Pre-PAR is more efficient than the previous state-of-the-art Meta-MGNN. Although Pre-GNN takes less time, its performance is much worse than the proposed Pre-PAR as shown in Table 2.

Table 4: Computational cost on Tox21.

| Method | # Episodes | Time (s) |
|---|---|---|
| ProtoNet | ~1800 | ~1065 |
| MAML | ~1900 | ~2388 |
| TPN | ~1800 | ~1274 |
| EGNN | ~1900 | ~1379 |
| PAR | ~1800 | ~2328 |
| Pre-GNN | ~1500 | ~491 |
| Meta-MGNN | ~1500 | ~1764 |
| Pre-PAR | ~1000 | ~1324 |

## C.2 Ablation Study on 1-shot Tasks

Figure 7 presents the results of comparing PAR (and Pre-PAR) with the seven variants (Section 4.3) on 1-shot tasks from Tox21. The conservation is consistent: PAR and Pre-PAR outperform these variants.

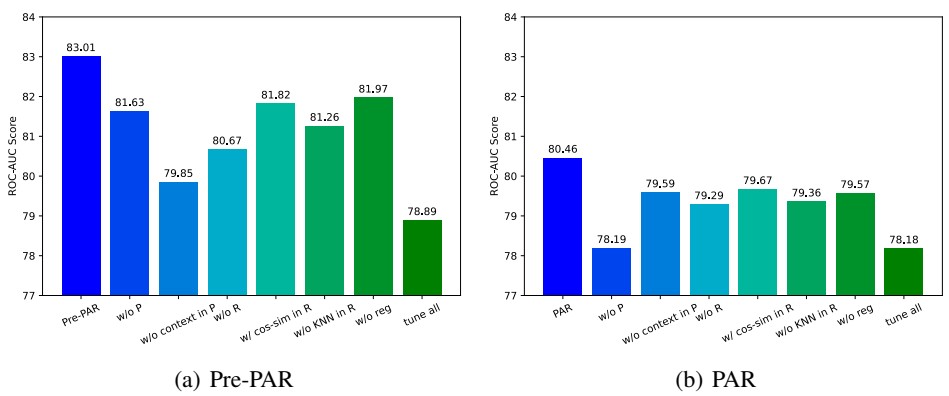

(a) Pre-PAR        (b) PAR

Figure 7: Ablation study on 1-shot tasks from Tox21.

## C.3 Details of Case Study

The details of the ten molecules used in Section 4.5 are presented in Table 5.

Table 5: The 10 molecules sampled from Tox21 dataset, which coexist in the three meta-testing tasks (the 10th task for SR-HSE, the 11th task for SR-MMP, and the 12th task for SR-p53).

| | Molecule | | Label | |
|---|---|---|---|---|
| ID | SMILES | SR-HSE | SR-MMP | SR-p53 |
| Mol-1 | Cc1cccc(/N=N/c2ccc(N(C)C)cc2)c1 | 0 | 1 | 0 |
| Mol-2 | O=C(c1ccccc1)C1CCC1 | 1 | 0 | 0 |
| Mol-3 | C=C(C)[C@H]1CN[C@H](C(=O)O)[C@H]1CC(=O)O | 0 | 0 | 1 |
| Mol-4 | c1ccc2sc(SNC3CCCCC3)nc2c1 | 1 | 1 | 0 |
| Mol-5 | C=CCSSCC=C | 0 | 0 | 1 |
| Mol-6 | CC(C)(C)c1cccc(C(C)(C)C)c1O | 0 | 1 | 0 |
| Mol-7 | C[C@@H]1CC2(OC3C[C@@]4(C)C5=CC[C@H]6C(C)(C)C(O[C@@H]7OC[C@@H](O)[C@H](O)[C@H]7O)CC[C@@]67C[C@@]57CC[C@]4(C)C31)OC(O)C1(C)OC21 | 0 | 1 | 0 |
| Mol-8 | O=C(CCCCCC(=O)Nc1ccccc1)NO | 0 | 0 | 1 |
| Mol-9 | CC/C=C\\C/C=C\\C/C=C\\CCCCCCC(=O)O | 1 | 0 | 0 |
| Mol-10 | Cl[Si](Cl)(c1ccccc1)c1ccccc1 | 0 | 1 | 0 |