# OpenReview forum: "Property-Aware Relation Networks for Few-Shot Molecular Property Prediction"
_NeurIPS.cc/2021/Conference — NeurIPS 2021 Spotlight_

### Official Review · Reviewer_8nqn · 2021-07-16

**Rating:** 6
**Confidence:** 4

**Summary:**

This paper proposes a property-aware adaptive relation networks for few shot molecular property prediction.  They propose a graph based molecule encoder which is trained from large scale tasks to capture a generic representation of the molecules and a property aware embedding function which adapts the generic molecule embedding to be property-aware. Next, an adaptive relation graph learning module is proposed to capture the relationship between similar molecules with respect to a target property. The adaptive relation graph learning module is based on GNNs. A new training strategy, where only the parameters of property aware embedding function and the classifier module is tuned, is also proposed. The authors argue that it useful to separately capture generic molecule representations and those that are specific to the particular task.

**Ethical Concerns:**

I don't see any ethical concerns with the paper.

**Limitations And Societal Impact:**

Limitations and Questions for authors:
Many of the molecular properties of interest are regression tasks (for example, quantum mechanical properties , binding affinities and so on). Can the authors explain if it is possible to extend the methodology to regression tasks?

The evaluation is done only on Tox21, Sider, MUV and Toxcast.    Does the method transfer to other tasks from the moleculenet dataset?

**Main Review:**

The authors propose a new method for Few shot learning of molecular properties. The proposed method achieves state of the art performance on four classification tasks (Tox21, Sider, MUV, ToxCast). As the authors argue, few shot learning for predicting molecular properties is very important for drug discovery and related applications. Collecting a large number of labels for a particular molecular property is usually expensive. Hence the proposed problem is of great practical interest.

The proposed method is novel and the experimentation is thorough, despite the evaluation being done only a small number of tasks (see question below about extending it to other tasks). The ablation study disentangles the contribution of each variant on the  final score. The paper is well written and clear.

Overall, I am quite positive about this paper. I feel it tackles an important problem in a novel way and shows good performance.

**Time Spent Reviewing:**

1.5

---

> ### Author Response · Authors · 2021-08-10
> **Response to Reviewer 8nqn**
>
> We would like to thank the reviewer for the positive comments!  For each detailed question, we provide response below.
>
> **Q1.** *Can the authors explain if it is possible to extend the methodology to regression tasks?*
>
> **A1.** Yes. A regression task estimates a regression function given K samples. It does not have classes nor prototypes. PAR and Pre-PAR can be extended to regression tasks with the following changes:
>
> - Property-aware embedding function: replace the two class prototypes in equation (2) by task embedding (which can be calculated as the average of all embeddings). In this way, property-aware molecular embeddings w.r.t the target property of the current task can still be obtained.
>
> - Adaptive Relation Graph: learn relation graph among molecules as before. As wrong neighbor cannot be easily decided like classification, one can choose not to sparsify the relation graph as a KNN graph or take K as a hyperparameter to tune.
>
> - Loss function in equation (6): replace classification loss (i.e., cross entropy) by regression loss such as root-mean-square error; neighbor alignment loss can be used as long as $\mathbf{A}^*_{T_\tau}$ is defined. For example, one can set $$[\mathbf{A}^*_{T_\tau}]\_{ij}=e^{-(y_{\tau,i}-y_{\tau,j})^2}$$ where $y_{\tau,i}$ and $y_{\tau,j}$ are the regression value for molecules $x_{\tau,i}$ and $x_{\tau,j}$ respectively.
>
> **Q2.** *The evaluation is done only on Tox21, Sider, MUV and ToxCast. Does the method transfer to other tasks from the moleculenet dataset?*
>
> **A2.** PAR and Pre-PAR can be applied to other tasks from MoleculeNet. MoleculeNet contains four types of datasets:
>
> - biophysics and physiology: these two types are already evaluated in the paper (Tox21, Sider, ToxCast belong to physiology while MUV belong to biophysics), which commonly tested in few-shot learning and transfer learning across datasets settings [1,4,16].
> - physical chemistry: it can be handled by GIN [19], therefore the current PAR and Pre-PAR can be directly applied.
> - quantum mechanics: it require modeling electronic properties. One can replace GIN by specially designed molecular encoder such as MGCN [b1] and DimeNet [b2] which may obtain better performance.
>
> If the tasks are regression, one also need to adapt PAR and Pre-PAR as described in **A1** above.
>
>
> [b1] Molecular property prediction: A multilevel quantum interactions modeling perspective. In Proceedings of the AAAI Conference on Artificial Intelligence, volume 33, pages 1052-1060, 2019.
>
> [b2] Directional message passing for molecular graphs. In International Conference on Learning Representations, 2019.

---

### Official Review · Reviewer_d5zA · 2021-07-16

**Rating:** 6
**Confidence:** 4

**Summary:**

This paper proposes a novel framework to solve a few-shot molecular property prediction problem w.r.t different properties. It takes GNN as the encoder to obtain property-aware molecular embeddings and projects it to a substructure-aware space according to the target property. Moreover, a relation graph between molecules is learned adaptively to refine the embeddings. Training strategies like gradient-based meta-learning is applied to boost property-aware training. Extensive experiments are conducted over a few-shot property prediction task as well as transfer learning across datasets

**Limitations And Societal Impact:**

See above

**Main Review:**

The authors design a novel graph-based property-aware framework for few-shot molecular property prediction and generate a relation graph among molecules to help further refine the molecular embeddings. However, with a closer look, the proposed model seems to be a combination of multiple existing works, e.g., property-aware attention[16], gradient-based meta-learning strategy [9]. It utilizes numbers of modules, and my concern is that how important each module is. It is hard to say all these modules/strategies work effectively since the performance improvement seems limited. The ablation study compares the performance of some variants but it only takes out one module at a time, so there is a chance that certain combinations can obtain competitive performance too.

In addition, the experiment results show that the performance of PAR does not significantly surpass other baseline models w/o pretrain models, instead the pre-GNN + PAR performs the best. The effectiveness of the proposed PAR seems unimpressive. Moreover, the details of the pretrain model implementation are not given, like what dataset is used and what experimental settings are. Also, I am curious that why the training takes so long (2000 epochs), while the original pre-GNN only trains 100 epochs. The dataset does not seem that large, it’s only a few thousand molecules. Additionally, considering such a large training epoch, it’s better to report the training time for the proposed model.

The paper seems to be finished in rush, many unfinished part, e.g., missing citation on P3 L101; no number filling out on P7 L249; the caption of Fig.5 is not consistent with the notation; the dataset description on P7 L216 is not consistent with table 1. The writing needs to be improved.


**Time Spent Reviewing:**

4 HOURS

---

> ### Author Response · Authors · 2021-08-10
> **Response to Reviewer d5zA**
>
> We would like to thank the reviewer for reviewing our paper. We are sorry that writing issues may prevent you from diving into our paper. We have proofread the paper strictly. For each detailed question, we provide response below.
>
> **Q1.** *You write in summary that "It takes GNN as the encoder to obtain property-aware molecular embeddings."*
>
> **A1.** GNN itself cannot obtain property-aware molecular embeddings.
>
> - As written on line 135-137, existing graph-based molecular encoders only capture generic knowledge. As a result, a molecule will obtain the same molecular embedding regardless of the target property considered in the current task.
>
> - However, a molecule can be evaluated for multiple properties (Figure 1). This leads to a one-to-many relationship between a molecule and properties, which makes molecular property prediction particularly hard.
>
> - Our proposed PAR makes the molecular embeddings property-aware by a specially designed function introduced in Section 3.2.
>
> **Q2.** *The proposed model seems to be a combination of multiple existing works, e.g., property-aware attention[16], gradient-based meta-learning strategy [9].*
>
> **A2.** Sorry for causing these misunderstandings. We elaborate the difference below.
>
> - Meta-MGNN [16] proposes task-aware attention (not property-aware attention): it calculates the average of sample embeddings as the task embedding which is then stored and used to weigh the contribution of each task in calculating the objective during meta-training.
> In contrast, we design property-aware embedding function which first co-adapts each molecule embedding with class prototypes and further projects it to a substructure-aware space w.r.t. the target property.
> Both the target and the methodology are different.
> - Gradient-based meta-learning strategy [9] (MAML): MAML updates the same set of parameters during meta-learning across different tasks and learning within each task.
> In contrast, as written on line 60-63 and 181-186, we propose a new training strategy where we only fine-tune the property-aware embedding function and final classifier while keeping the other parts (graph-based molecular encoder and adaptive relation graph learning module) fixed within each task.
> This design can help to separately capture the generic knowledge shared across different tasks and those property-aware. In ablation study, we compare PAR with variant-6 which corresponds to fine-tune all parameters by gradient descent as in MAML [9]. Results in Figure 3 and Figure 5 show our new training strategy can significantly improve the performance.
>
> Our method is proposed in particular for molecular property prediction tasks. We design three key components (line 55-63):
>
> 1. property-aware embedding function to transform the generic molecular embeddings to substructure-aware space relevant to the target property;
> 2. adaptive relation graph learning module to jointly estimate molecular relation graph and refine molecular embeddings w.r.t. the target property; and  a new neighbor alignment loss to encourage the relation graph to choose correct neighbors;
> 3. a new training strategy to encourage the model to capture generic and property-aware information separately.
>
> None of these have been investigated in existing methods. Please kindly refer to R1' comments.
>
> **Q3.** *The experiment results show that the performance of PAR does not significantly surpass other baseline models w/o pretrain models, instead the pre-GNN + PAR performs the best. The effectiveness of the proposed PAR seems unimpressive.*
>
> **A3.**
>
> 1. Please note that both graph-based molecular encoders learned from scratch or pretrained are useful:
>    although pretrained graph-based encoder encodes rich generic molecular information by learning enormous unlabeled data, encoders learning from scratch can carry some new insights. For example, the recently proposed DimeNet [a1] is able to model directional information such as bond angles and rotations between atoms, which has no pretrained version. As our proposed method can use any molecular encoder to obtain generic molecular embedding, it can easily accommodate any newly proposed molecular encoder w/o or w/ pretraining.
>
> 2. In fact, PAR obtains the best performance among methods using graph-based molecular encoders learned from scratch, while Pre-PAR obtains the best among methods using pretrained graph-based molecular encoders.
> To see this clearer, the table below summarizes the average improvement of PAR and Pre-PAR based on results reported in Table 2 and Table 5.
>
> | relative improvement ratio                    | Tox21 |        | SIDER |       | MUV   |       | ToxCast |       | Average |
> | --------------------------------------------- | ----- | ------ | ----- | ----- | ----- | ----- | ------- | ----- | ------- |
> | n-shot                                        | 10    | 1      | 10    | 1     | 10    | 1     | 10      | 1     |         |
> | PAR over the best baseline w/o pretrained GIN | 1.05% | -0.62% | 2.48% | 0.20% | 0.91% | 3.12% | 4.39%   | 1.99% | 1.59%   |
> | PAR over the best baseline w pretrained GIN   | 2.36% | 1.07%  | 3.51% | 1.50% | 1.41% | 2.14% | 1.95%   | 1.00% | 1.49%   |
>
> As shown,  PAR obtains significant better performance than the best baseline learned from scratch methods (e.g. EGNN) by 1.59%, and Pre-PAR is better than the b pretrained methods (e.g. Meta-MGNN) by 1.49%.
>
> In the revised version, we will separately highlight the best performance among methods using graph-based molecular encoders learned from scratch and methods using pretrained graph-based molecular encoders in Table 2,3,5.
>
> [a1]. Directional message passing for molecular graphs. In International Conference on Learning Representations, 2019.
>
> **Q4.** *My concern is that how important each module is. It is hard to say all these modules/strategies work effectively. The ablation study compares the performance of some variants but it only takes out one module at a time, so there is a chance that certain combinations can obtain competitive performance too.*
>
> **A4.** The 6 variants considered in ablation study can be grouped according to their evaluation target:
>
> - Evaluate the design of property-aware embedding function: (variant-1) w/o property-aware embedding, and (variant-2) w/o context $\mathbf{e}^c_{x_{\tau,i}}$ in equation (2);
> - Evaluate the design of adaptive relation graph learning module: (variant-3) w/o adaptive relation graph learning, (variant-4) w/o reducing $\mathcal{G}_{T_\tau}$ to KNN graph, and (variant-5) w/o the neighbor alignment loss in equation (6);
> - Evaluate the effectiveness of our training strategy: (variant-6) fine-tune all parameters on line 15 of Algorithm 1.
>
> These variants cover all components of training PAR, without overlapping functionalities. As shown in Figure 3 and Figure 6, the removal of any component leads to significant performance drop. Therefore, there is no chance that other combination of these components will obtain competitive results.
>
> Both R1 and R3 acknowledge the value of our ablation study.
>
> **Q5.** *The details of the pretrain model implementation are not given.*
>
> **A5.** On line 225, we mention PAR uses Pre-GNN[19]. We provide details of Pre-GNN on line 472-475 in Appendix A. In footnote 8, we provide the link to downloaded the Pre-GNN and refer the details to Appendix A of [19].
>
> **Q6.** *why the training takes so long (2000 epochs), while the original pre-GNN only trains 100 epochs. considering such a large training epoch, it’s better to report the training time for the proposed model.*
>
> **A6.** Thanks for pointing this out. We should use **episode** instead of **epoch** on line 239 as in [20,27] as we conduct episodic training. In PAR, episode corresponds to the times of repeating the while loop (line 2-19) in Algorithm 1.
>
> Besides, Pre-GNN is trained on ZINC15 dataset, whose epoch number does not apply for datasets used in this paper.  The results of 10-shot learning on Tox21 dataset are summarized below.
>
> |   Method   | Time per episode (s) | # episodes | Training time (s) |
> | :--------: | :------------------: | :--------: | :---------------: |
> |  ProtoNet  |        0.592         |   ~1800    |       ~1065       |
> |    TPN     |        0.708         |   ~1800    |       ~1274       |
> |    MAML    |        1.257         |   ~1900    |       ~2388       |
> |    EGNN    |        0.726         |   ~1900    |       ~1379       |
> |    PAR     |        1.324         |   ~1800    |       ~2328       |
> |  Pre-GNN   |        0.327         |   ~1500    |       ~491        |
> | Meta-MGNN  |        1.176         |   ~1500    |       ~1764       |
> |  Pre-PAR   |        1.324         |   ~1000    |       ~1324       |
>
> As shown, Pre-PAR is more efficient than the current state-of-the-art Meta-MGNN. Although Pre-GNN takes less time, its performance is much worse than Pre-PAR as shown in Table 2,3,5.
>
>
> **Q7.** *the dataset description on P7 L216 is not consistent with table 1.*
>
> **A7.** They are indeed consistent. The original 5868 side effect categories for 1427 compounds are grouped into 27 categories which correspond to 27 tasks in Table 1, which follows [4,16]. Please check description of SIDER on page 6 of [4].

---

> > ### Comment · Area_Chair_ouFF · 2021-08-30
> > **Acknowledging Authors rebuttal**
> >
> > Dear Reviewer thank you for your review, authors have provided their answer, as we are approaching decision phase could you please check if they addressed your concerns.

---

> > > ### Comment · Reviewer_d5zA · 2021-08-31
> > > **Response!**
> > >
> > > Thanks for the thoughtful and thorough response. Most of my concerns have been clarified. I am happy to increase my score to 6.
> > >
> > > 1. The claim: “GNN itself cannot obtain property-aware molecular embeddings”seems not correct. The GNN can always gain some property-aware information by giving some supervised training procedures.
> > >
> > > 2. Regarding the novelty, I still feel that the novelty of this paper is not significant. It is more like an application of the meta-learning technique to the molecular property prediction task.
> > >
> > > 3. It is claimed that the obtained embedding can capture property-aware substructures, but no strong evidence like molecular graph structure/substructure visualization proves such assumption. Proposed property-aware embedding function may generate the different embeddings according to target property, but it may not capture the substructures that affect the property.
> > >
> > > 4. Some mistakes in the original paper which makes it somehow difficult to read, e.g., the last two terms in eq(3) are the same.
> > >
> > > It is strongly suggested that the authors add more elaborations about why this framework is suited for this task in the revised version.

---

> > > > ### Author Response · Authors · 2021-09-01
> > > > **Thank you for supporting us now**
> > > >
> > > > Dear reviewer, we are grateful that you now vote for us. We will rewrite and proofread the paper based on your suggestions. Thanks for your time and efforts!

---

### Official Review · Reviewer_VWYi · 2021-07-27

**Rating:** 8
**Confidence:** 4

**Summary:**

The authors look at molecular property prediction from a few shot learning perspective. They propose an approach, closely following Meta-MGNN, to adapt part of a (optionally pre-trained) graph embedding network in MAML-style per task. A number of well-motivated novelties are introduced: (a) adapt part of the GNN encoder, called property-aware transformation (vs just task-specific attention in Meta-MGNN). (b) explicitly constructing the similarity graph between a test sample and the support set (labeled samples) and add a sparsity loss for the adjacency matrix during training. The results, focusing on 1-shot and 10-shot prediction, convincingly show gains in this regime, and extensive ablations and baselines are presented.


**Limitations And Societal Impact:**

It would be helpful to see a section on limitations.
Negative societal impact: N/A

**Main Review:**


Major strengths:



* Strong motivation for property-specific adaptation + adaptive relation graph learning is presented.  This is done via examples (Fig1), comparisons to works without property-specific adaptation, and examples of how learned relation graphs differ greatly across tasks.
* The proposed model (PAR), though complex, seems to appropriately address the problem setup.
* Strong results (only on 1-shot and 10-shot classification) + comparison to multiple baselines, for each of PAR’s major dimensions.
* Comprehensive ablations of contributions and analysis of their effects.

Weaknesses - none are major, all are expanded upon below.



* Somewhat unclear writing.
* High overall complexity of the approach, adding elements to already-complex IterRefLSTM and Meta-MGNN
* Results only on artificial few-shot settings

Overall this is a strong paper, hence a score of 8: Top 50% of accepted papers, clear accept

## Detailed review


### Motivation and method

The paper has a strong motivation around few-shot learning and the need for property-specific adaptation of general purpose embeddings.

Also the motivation behind ensuring the relation graph (similarity  is correctly constructed for a task is well-taken, and illustrated convincingly with Fig 4,8,9. Ablations show the gain of constructing and sparsifying the similarity graph.

Questions for the authors:



* Can the adjacency matrix be estimated directly from cosine similarity of embeddings, rather than doing the iterative estimation?

Other minor comments:



* Appendix Figure 8 row order doesn’t seem to match caption? (0-1 should be ground truth versus grayscale predicted?)
* Figure 4 - these adjacency matrices are the same but according to Fig9 and caption they should be different. Also same comment as above.
* Figure 3 - please label the variants with a short summary to avoid having to cross-reference the text.


### Impact / significance / results



* The ablations are strong.  The three main contributions (beyond results) are ablated, and show large degradations when removed:
    * property-aware embedding
    * fine-tuning of all parameters degrades wrt only final layers
* Comparison to existing work is thorough.
    * 10 seeds for confidence intervals.  Proper comparison to several lines of work which were the basis for PAR.
* Overall this paper addresses a difficult task in a better way, with strong motivation and appropriate novel methods.

Comments:



* It would be helpful to see a section on limitations.
* It would be interesting to see results on full tox21 / other moleculenet datasets, without artificially constraining to 1 or 10-shot classification. However we recognize this is following the standard set by IterRefLSTM and Meta-MGNN.
* See suggested ablation above: “Can the adjacency matrix be estimated directly from cosine similarity of embeddings, rather than doing the iterative estimation?”


### Clarity of writing

Overall the paper is thorough, well-motivated and contains necessary detail.

The **clarity of the paper could be improved**:



* Some concepts should be high-level described (relation graph / property-aware embedding) when the terms are introduced.
* Heavy notation can be reduced and gets in the way of understanding. Authors should look for places to provide a high level summary sentence at the start of the paragraph to summarize the paragraph.
* It would be helpful to be more explicit where the aggregation (GNN node embeddings -> single vector) happens, versus where is task-specific adaptation.

Specific comments about equations:



* Please move description of g, p, r, c superscripts to the front of [Section 3 - Proposed Method]
    * ‘c’ superscript is used for both classifier parameters $\theta^c$ and context embedding $e^c$, which occur in two different stages?
* What is the $e^p_{x_{Tj}}$ initialization value for equation (2)?

Minor comments and typos

* Several grammatical errors (eg: “while fine-tune theta”, “taking a few gradient descents”)
* Figure 4: (d) 12th task instead of 11th task?
* Parameterize MLP in eq (3) by $\theta^r$ ?
* Also see comments about figures above

**Time Spent Reviewing:**

10

---

> ### Author Response · Authors · 2021-08-10
> **Response to Reviewer VWYi**
>
> We would like to greatly thank the reviewer for the positive comments and valuable time spent on reviewing our paper!  We are also grateful for the constructive suggestions on improving the clarity of the paper. We will revise the paper accordingly. For each detailed question, we provide response below.
>
> **Q1.** *High overall complexity of the approach, adding elements to already-complex IterRefLSTM and Meta-MGNN.*
>
> **A1.** PAR is not more complex than IterRefLSTM and Meta-MGNN.
> 1. Please note that our PAR is not incremental upon IterRefLSTM nor Meta-MGNN.
>     Although all three methods use graph-based molecular encoder, the rest components of respective models are quite different (line 37-42):
>   - IterRefLSTM designs the Iterative Refinement LSTMs,
>   - Meta-MGNN constructs self-supervised tasks and calculates task embedding which is then used to weigh the contribution of each task in meta-training,
>   - Our PAR contains property-aware embedding function and adaptive relation graph learning module.
>
> 2. To compare the computational cost empirically, we compare Pre-PAR with Meta-MGNN which both use pretrained graph-based molecular encoders.
> We record the training time and training episodes which corresponds to the times of repeating the while loop (line 2-19) in Algorithm 1. IterRefLSTM is not compared due to the lack of public codes. The results of 10-shot learning on Tox21 dataset are summarized below.
>
> |   Method   | Time per episode (s) | # episodes | Training time (s) |
> | :--------: | :------------------: | :--------: | :---------------: |
> | META-MGNN  |        1.176         |   ~1500    |       ~1764       |
> |  Pre-PAR   |        1.324         |   ~1000    |       ~1324       |
>
> As shown, Pre-PAR takes less time and episodes to converge. As for space requirement, as written on line 485 of Appendix B.1, Meta-MGNN runs out of memory on the large dataset ToxCast.
>
> **Q2.** *Can the adjacency matrix be estimated directly from cosine similarity of embeddings, rather than doing the iterative estimation?*
>
> **A2.** Yes, it can but it may not be able to model relation graphs among molecules properly.
>
> 1. The current adjacency matrix is estimated via a learned similarity function. As suggested, we now add a baseline: replace equation (3) by obtaining adjacency matrix by the fixed cosine similarity, which is calculated as $$[\mathbf{A}^{(l)}_{T\_\tau}]\_{ij}=\frac{(\mathbf{h}^{(l-1)}\_{\tau,i})^\top\mathbf{h}^{(l-1)}\_{\tau,j}}{\|\mathbf{h}^{(l-1)}\_{\tau,i}\|\|\mathbf{h}^{(l-1)}\_{\tau,j}\|},$$ and keep the rest the same as PAR. Thus, this baseline also refines molecular embeddings on the relation graph constructed by cosine similarity. The results are summarized below.
>
>    As shown, PAR obtains much better performance.  This validates the necessity of learning a similarity function from the data rather than using the fixed cosine similarity. We will add this new baseline into Section 4.3.
>
> | Method  | Adjacency         | #itertaion $L$(line 9 in Alg.1) |  10-shot  |  1-shot   |
> | ------- | ----------------- | :-----------------------------: | :-------: | :-------: |
> | PAR     | cosine similarity |                1                |   81.12   |   79.67   |
> |         |                   |                2                |   80.85   |   78.36   |
> |         |                   |                3                |   80.07   |   76.44   |
> |         | equation (3)      |                1                |   81.78   |   80.02   |
> |         |                   |        **2 (reported)**         | **82.06** | **80.49** |
> |         |                   |                3                |   81.11   |   77.85   |
> | Pre-PAR | cosine similarity |                1                |   83.47   |   81.82   |
> |         |                   |                2                |   83.1    |   81.53   |
> |         |                   |                3                |   82.67   |   81.08   |
> |         | equation (3)      |                1                |   83.81   |   82.12   |
> |         |                   |        **2 (reported)**         | **84.93** | **83.01** |
> |         |                   |                3                |   84.26   |   82.44   |
>
> 2. Please also note that the parameter $\mathbf{\theta}^r$ of the adaptive relation graph learning module includes both the parameter of MLP in equation (3) and parameter of GNN in equation (4). For a new task at meta-testing stage, $\mathbf{\theta}^r$ is fixed and directly used.
>
> Thus, the use of adaptive relation graph molecule is not expensive.
>
>
> **Q3.** *It would be helpful to see a section on limitations.*
>
> **A3.** Thanks for the suggestion, we will add it in the paper: we leave interpreting the substructures encoded in the property-aware molecular embeddings as future works, such as connecting them to functional groups.
>
>
> **Q4.** *It would be interesting to see results on full tox21 / other moleculenet datasets, without artificially constraining to 1 or 10-shot classification. However we recognize this is following the standard set by IterRefLSTM and Meta-MGNN.*
>
> **A4.** Thanks for the suggestion. We design PAR to handle molecular property prediction tasks under few-shot learning setting. But it indeed can work on full moleculenet datasets. Following the setting of [19], we now conduct experiments on the full Tox21 dataset and summarize the results below.
> PAR takes GIN as the generic molecular encoder.
>
> | ROC-AUC score      | Learned from scratch | Pretrained |
> | ------------------ | -------------------- | ---------- |
> | GIN                | 74.0                 | 78.1       |
> | PAR(GIN as encoder)| 75.1                 | 79.6       |
>
> As shown, the design of property-aware embedding function, adaptive relation graph learning module, and the neighbor alignment loss are helpful to improve the performance of GIN on standard supervised learning.
>
>
> **Q5.** *What is the $\mathbf{e}^p_{x_{\tau,i}}$ initialization value for equation(2)?*
>
> **A5.** $\mathbf{e}^p_{x_{\tau,i}}$ is transformed from $\mathbf{e}^g_{x_{\tau,i}}$, which does not need initialization. The confusion may come from our typo, we now rewrite line 137-139 as:
> "Let $\mathbf{p}\_c$ denotes the class prototype for class $c\in\{0,1\}$, which is computed as the average of $\mathbf{e}^g\_{x\_{\tau,i}}$ in $\mathcal{S}\_\tau$ whose $y\_{\tau,i}=c$.
> We model the context for $x\_{\tau,i}$ as $\mathbf{C}\_{\tau,i}=[{(\mathbf{e}^g\_{x\_{\tau,i}})}^\top;\mathbf{p}^\top\_0;\mathbf{p}^\top\_1]\in\mathbb{R}^{3\times d^g}$."
>
>
> **Q6.** *superscript is used for both classifier parameter $\mathbf{\theta}^c$ and context embedding $\mathbf{e}^c$, which occur in two different stages?*
>
> **A6.** We will use 'b' superscript to denote context instead.
>
> **Q7.** *Minor comments and typos.*
>
> **A7.** Thank you for the advices on improving the clarity of this paper. We will revise the paper accordingly.
>
> - In Figure 4, we wrongly paste adjacency matrices for the 11th task twice. Please refer to Figure 8 in Appendix B.2.3. In Figure 8, the 1st row is $\mathbf{A}_{T\_\tau}^*$ calculated using ground-truth labels and the 2nd row is adjacency matrix learned by PAR.
> - Figure 5 (d) should be 12th task.
> - We will label the variants in Figure 3 as below.
>
> | old-name  | new-name | description                                                  |
> | --------- | -------- | ------------------------------------------------------------ |
> | variant-1 | P-emb    | w/o property-aware embedding                                 |
> | variant-2 | P-ctx    | w/o context in equation (2)                                  |
> | variant-3 | R-g      | w/o adaptive relation graph                                  |
> | variant-4 | R-knn    | w/o KNN selection in relation graph                          |
> | variant-5 | L-adj    | w/o the neighbor (adjacency) alignment loss in equation (6)  |
> | variant-6 | tuneall  | fine-tune all parameters                                     |

---

> > ### Comment · Reviewer_VWYi · 2021-08-30
> > **reply to authors response**
> >
> > I thank the authors for their in-depth response.
> > The additional results further strengthen the value of one of the core contributions beyond few-shot, and iterative adjacency matrix estimation.
> > -- though I want to remark it is weird to see degradation with L>2!
> >
> > I still strongly suggest to accept this paper (maintain rating of 8).
> >
> > I further want to re-iterate my advice to the authors to rewrite the paper thoroughly for camera ready version, look for reduced notational complexity and moving things to appendix will make the core ideas be more salient and reach higher impact with a wider audience.

---

> > > ### Author Response · Authors · 2021-08-30
> > > **Thank you for supporting us**
> > >
> > > We sincerely thank the reviewer for supporting us!
> > >
> > > We are happy to see our additional results help. A larger L may lead to oversmooth, while a small L may not be able to obtain expressive representations. In this paper, we treat L as a hyperparameter.
> > >
> > > We will make the camera ready version as clear as possible, carefully rewriting the paper following the reviewers' advices.
> > > Thanks for the advice and time again!

---

### Decision · Program_Chairs · 2021-09-27

**Decision:**

Accept (Spotlight)

**Comment:**

The paper proposes  a framework for few shot learning for molecular properties prediction. Reviewers have agreed that the problem of few shot training is important in the molecular  space and that the  approach proposed in the paper is novel and elegant. Authors showed evidence of their method working in the few shot regime. Accept